Antimicrobial activity of Lactobacillus spp. isolated from fermented foods and their inhibitory effect against foodborne pathogens

Hussein Athraa Oudah 1 2
Khalil Khalida 1
Mohd Zaini Nurul Aqilah 3
Al Atya Ahmed Khassaf 2
Aqma Wan Syaidatul syaidatul@ukm.edu.my 1
1 Department of Biological Sciences and Biotechnology, Faculty of Science and Technology, Universiti Kebangsaan Malaysia , Bangi , Selangor , Malaysia
2 Department of Biology, Faculty of Science, Thi Qar University , Thi Qar , Iraq
3 Department of Food Sciences, Faculty of Science and Technology, Universiti Kebangsaan Malaysia , Bangi , Selangor , Malaysia
Brygadyrenko Viktor
Electronic publication date: 2025 Jan 6
Publication date: 2025
Volume: 13
Electronic Location ID: e18541
Received 2024 Sep 9; Accepted 2024 Oct 27
Copyright: ©2025 Hussein et al.
Copyright year: 2025
Copyright holder: Hussein et al.
License: This is an open access article distributed under the terms of the Creative Commons Attribution License, which permits unrestricted use, distribution, reproduction and adaptation in any medium and for any purpose provided that it is properly attributed. For attribution, the original author(s), title, publication source (PeerJ) and either DOI or URL of the article must be cited.
License URL: https://creativecommons.org/licenses/by/4.0/

Keywords: Lactobacillus spp., Antibiofilm, Fermented foods, Foodborne pathogens, L. plantarum KR-3 CFS, Pathogens, Antimicrobial

Funding: Universiti Kebangsaan Malaysia under a Research University Grant GUP-2023-075 Financial support was provided by Universiti Kebangsaan Malaysia under a Research University Grant (GUP-2023-075). The funders had no role in study design, data collection and analysis, decision to publish, or preparation of the manuscript.

==============================
Lactic acid bacteria (LAB), known for their health benefits, exhibit antimicrobial and antibiofilm properties. This study investigated the cell-free supernatant (CFS) of Lactobacillus spp., particularly L. plantarum KR3, against the common foodborne pathogens S. aureus, E. coli and Salmonella spp. Lactobacillus strains were isolated from cheese, pickles and yoghurt. They were then identified by morphological, physiological and biochemical characteristics and confirmed by 16S rRNA gene sequencing. Culture supernatants from seven lactobacilli isolates showed varying inhibitory activities. Notably, L. plantarum KR3 and L. pentosus had the highest bacteriocin gene counts. L. plantarum KR3 CFS demonstrated significant antibacterial activity, with inhibition zones of 20 ± 0.34 mm for S. aureus, 23 ± 1.64 mm for E. coli, and 17.1 ± 1.70 mm for Salmonella spp. The CFS also exhibited substantial antibiofilm activity, with 59.12 ± 0.03% against S. aureus, 83.50 ± 0.01% against E. coli, and 60. ± 0.04% against Salmonella spp., which were enhanced at the minimum inhibitory concentration (MIC). These results highlighted the potential of L. plantarum KR3 in antimicrobial applications, however, further research is needed to evaluate its viability and functional properties for probiotic use. Additionally, the CFS demonstrated exceptional thermal stability, reinforcing its promise as an antimicrobial agent.

Introduction

Lactic acid bacteria (LAB) are crucial in food fermentation are commonly used to produce yoghurt, cheese, cultured butter, sour cream, sausages, pickles, olives, sauerkraut and cocoa (Kazou et al., 2021). LAB help preserve natural foods and restore the biodiversity lost during pasteurisation, thereby enhancing the flavour and speeding up maturation (Bintsis, 2018a). Among them, Lactobacillus plantarum, used to ferment meat and wine, is significant in the dairy industry (Brizuela et al., 2018). Known as probiotics, LAB are gaining interest for their health benefits, balancing intestinal microbial populations and protecting against pathogens. LAB are vital in food processing and fermentation, and are recognised for their safe status (GRAS) and biopreservative potential (Ngene et al., 2019).

LAB exert strong antagonistic effects on food-contaminating microorganisms by producing antimicrobial metabolites like organic acids (lactate, acetate and butyrate), hydrogen peroxide and bacteriocins, competing with harmful bacteria for nutrients or adhesion sites (Yaacob et al., 2022). Studies on LAB’s antibacterial properties in fermented foods emphasise their role in shelf-life extension, organoleptic qualities and nutritional quality through organic acid production. The strains include Lactiplantibacillus plantarum, Lacticaseibacillus paracasei, Lactiplantibacillus plantarum, Lactiplantibacillus plantarum, Lacticaseibacillus paracasei, Lactiplantibacillus pentosus and Lacticaseibacillus paracasei. Bacteriocins, ribosomally produced peptides or proteins, show antimicrobial activity against various strains. Advanced bacteriocins targeting foodborne pathogens are eliciting research interest for their cost effective and safe biopreservative potential. They slow the spread of pathogens like L. monocytogenes, E. coli, S. aureus and Salmonella spp. (Atieh et al., 2021).

The present study aims to isolate Lactobacillus spp. from cheese, pickles, and yogurt, evaluate the production of cell-free supernatants (CFS), and determine their antimicrobial spectrum against foodborne pathogens, including their antibiofilm activity. Additionally, the inhibition patterns of these CFS will be examined under different temperature, pH, and enzyme conditions. The results are expected to provide new insights into developing biopreservative agents to prevent and control pathogenic bacterial infections.

Materials & Methods

Sampling and isolation of Lactobacillus spp. from fermented foods

Fermented food samples including cheese, pickles and yoghurt were collected from various markets in April 2021. Samples were transported under sterile conditions and stored at 4 °C. Each sample (1 g) was dissolved in 9 mL of MRS broth and blended using a Stomacher machine (Stomacher 400, Seward, West Sussex, UK) for 5 min. The inoculated tubes were incubated at 37 °C for 48 h under anaerobic conditions. Tenfold serial dilutions were conducted, and subculturing was performed on solid MRS medium. After incubation, prominent white colonies were purified on MRS agar plates. Identification involved Gram staining, selective agar growth and biochemical tests. Acid production ability was tested on MRS. Only Gram positive, catalase, and oxidase-negative isolates were considered potential Lactobacillus spp. candidates. Stock cultures were kept at −20 °C in 15% (v/v) glycerol. Molecular identification of all isolated strains from each sample was carried out following the procedure of Wang et al. (2020). Genomic DNA was extracted from all bacterial isolates using a DNA extraction kit (Abcam, Cambridge, UK) according to the manufacturer’s protocol. The quantity of extracted DNA was measured using a Nanodrop One spectrophotometer (Thermo Scientific, Waltham, MA, USA). This DNA was used as the template for PCR amplification of the 16S rRNA gene sequences from each strain. The primers for amplification, 27F (5′-AGAGTTTGATCCTGGCTCAG-3′) and 1492R (5′-TACGGTTACCTTGTTACGACTT-3′, that amplified a region of approximately 1,500 bp. The PCR reaction used 25 µL of OneTaq Quick-Load 2X Master Mix with Standard Buffer (New England BioLabs, Ipswich, MA, USA), 1 µL of forward primer at 10 pmols/µL, 1 µL of reverse primer at 10 pmols/ µL, 1.5 µL of DNA template, and 9 µL of nuclease-free water. PCR was performed using a (Biometra TPersonal thermal cycler, Analytik, Jena, Germany) under the following conditions: initial denaturation at 94 °C for 3 min, followed by 30 cycles of denaturation at 94 °C for 30 s, annealing at 60 °C for 30 s, and extension at 72 °C for 2 min, with a final extension at 72 °C for 10 min. The resulting PCR products were confirmed by agarose gel electrophoresis, purified and sequenced. Sequence identity was determined using BLAST (Nucleotide BLAST, 16S rRNA sequence database). Electrophoresis was performed at 80 volts for 45 h on a 1.5% agarose gel stained with FloroSafe DNA Stain (1 µL/mL 1st BASE, Singapore). The following strains were identified from various fermented food samples: Lactiplantibacillus plantarum from yoghurt, Lactobacillus paracasei from feta cheese, Lactiplantibacillus plantarum from kafir cheese, Lactiplantibacillus plantarum from Syrian cheese, Lactobacillus paracasei from Turkish cheese, Lactiplantibacillus pentosus from mixed pickles, and Lactobacillus paracasei from green olive pickles.

Screening of Lactobacillus spp. for antagonistic activity

Preparation of cell-free supernatant extracts

Each of the seven active isolates was individually inoculated at 1% (v/v) into 50 mL of sterile MRS broth (Hangzhou Microbial Reagent) and incubated overnight at 37 °C for 24 h. CFS were obtained by centrifugation at 8,000 rpm for 15 min at 4 °C and the supernatant was passed through 0.22 µm filter membrane to ensure the complete removal of any remaining cells and pH was adjusted to 6.5 by using 1 M NaOH (Wayah & Philip, 2018). Enzymatic treatments with proteinase K, pepsin and lysozyme were then performed on the CFS to assess the proteinaceous nature of the bacteriocins.

Well-diffusion method

According to Clinical and Laboratory Standards Institute protocol, Mueller Hinton agar (MHA) and brain–heart infusion (BHI) broth were utilised as the standard media. The indicator strains E. coli, S. aureus, and Salmonella spp. isolated from food samples at UKM’s Department of Food Science, Universiti Kebangsaan Malaysia (UKM) were cultured on these media for 24 h at 37 °C. Under aerobic conditions, MHA was prepared, poured into sterile Petri dishes, and allowed to solidify. A standardised bacterial suspension was prepared by dissolving the bacteria in tubes containing sterile distilled water. Using a sterile swab stick, the organisms was spread on the surface of the MHA agar plates. Using a sterile blue tip, an 8 mm-diameter well was carefully created in the compacted agar, and 200 µL of supernatant culture was added into the well. The plates were cooled for 2 h. After 24 h of incubation at 37 °C, the plates were measured to determine the extent of zone of inhibition, indicating the inhibitory effects of the culture on the surrounding area (Abisado et al., 2018).

Overlay-plate method

The plates were first filled with 20 mL of MHA medium as the lower layer of medium. Then, the three indicator pathogenic bacteria that had been cultured overnight were added to (5 mL) of MHA soft agar, which had been cooled to about 50 °C with a 3% inoculum. Once properly combined, they were placed on the plate instantly to solidify. The sterilised Oxford cups were placed on the MHA agar surface and lightly pressed. Finally, sterile MHB broth was added to the control group, and (200 µL) of the isolated LAB CFS was poured into the cups. After 24 h of incubation at 37 °C, the plates were measured to determine the extent of zone of inhibition, indicating the inhibitory effects of the culture on the surrounding area (Muhammad et al., 2019).

Detection of bacteriocin synthesis genes in Lactobacillus spp.

The presence of 10 bacteriocin-related genes was investigated in seven Lactobacillus isolates by PCR using specific primers. The PCR primers and annealing temperatures are presented in Table 1. PCR conditions were similar for all genes: an initial denaturation step at 94 °C for 5 min, followed by 34 cycles of denaturation at 94 °C for 30 s, an extension at 68 °C for 60 s, and a final extension at 68 °C for 10 min (Macwanaa & Muriana, 2012). The PCR products were identified by electrophoresis on 1.5% agarose gels stained with FloroSafe DNA Stain.

Effects of temperature, pH, and enzymes on L. plantarum KR-3 CFS

To analyse pH tolerance, L. plantarum KR-3 CFS was adjusted to pH 2.0, 4.0, 6.0, 8.0, and 10.0 by using HCl (3 mol/L) or NaOH (3 mol/L). After 1 h at 37 °C, samples were readjusted to their initial pH. For thermal stability, L. plantarum KR-3 CFS was exposed to 25 °C, 40 °C, 60 °C, 80 °C, and 100 °C for 1 h. Enzyme treatments involved adding proteinase K, pepsin, and lysozyme (1 mg/mL) into the CFS, incubating at 37 °C for 2 h, and inactivating at 80 °C for 10 min. Untreated CFS served as the control. Antibacterial efficacy against bacterial pathogens was assessed using the Oxford cup double-layer plate method. This experiment was repeated three times (Hwanhlem, Chobert & H-Kittikun, 2014).

Biofilm eradication concentration determination method

The efficacy of L. plantarum KR3 CFS against biofilm formation was assessed in 96-well plates (Nest Biotechnology, Wuxi, China) following the method outlined by Jiang et al. (2022). L. plantaraum KR-3CFS was diluted to concentrations of 1 × MIC and 2 × MIC, and cultures of S. aureus, E. coli, and Salmonella spp. at a concentration of (107 CFU/mL) were added to the microplates containing various dilutions of CFS. Each well of the 96-well microtiter plate was filled with 100 µL of bacterial suspension and 100 µL of CFS, resulting the total volume in each well to 200 µ. Negative control wells were filled with 200 µL of media only. After mixing, the microplates were incubated for 24 h at 37 °C, the supernatant was removed, and the residue was washed with PBS. The biofilm biomass was stained with 0.1% crystal violet solution. Following a 15 min incubation period at room temperature, the plates were washed and allowed to air dry, followed by another washing step to remove excess dye. The retained crystal violet, indicating biofilm biomass, was solubilized with acetic acid and the absorbance was measured at 595 nm using a microplate reader (Varioskan LUX, Thermo Fisher Scientific, Waltham, MA, USA). The experiment was performed in triplicate.

Scanning electron microscopy analysis

Approximately 3 mL of S. aureus, E. coli, and Salmonella spp. culture, each with a concentration of 107 CFU/mL was centrifuged at 7,104× g at 4 °C for 3 min. Afterward, the harvested cells were washed once with PBS buffer and then resuspended in 1 mL of L. plantaruum KR3 at 37 °C for 2 h. The treated bacterial cells were fixed with 2.0% glutaraldehyde at 4 °C for 8 h and dehydrated in a series of ethanol solutions with increased concentration (30% to 100%) for 15 min each. The samples were then transferred onto polished silicon wafers measuring 10 × 10 mm2 and left to dry at 25 °C. After drying, the samples were coated with gold powder before imaging under a scanning electron microscope (JEOLJSM 6510 IV SEM). Untreated S. aureus, E. coli, and Salmonella spp. cells served as the control. Each experimental condition was performed in triplicate (Li et al., 2015).

Statistical analysis

Data analysis was performed, and results are presented as the mean ± standard deviation of three independent replicates. One-way ANOVA was conducted to assess significance levels among multiple groups. A P value < 0.01 was considered significant.

Table 1 Evaluating antimicrobial-produce of Lactobacillus spp. isolated from fermented foods and their inhibitory effect against foodborne pathogens.

Provides PCR primers and annealing temperatures for detecting 10 bacteriocin-related genes in seven Lactobacillus isolates.

Genes	Primers	References	
	Forward	Reverse	Length (bp)		
plnA	AAG AGT AGT GCG TATT	TTA CCA TCC CCA TTT TTT A	114	Wu et al. (2021)	
plnB	GCG AAA CAG TGC GGG GTT A	TGC TGT TCA ACA AGA TCC CGC	308	Wu et al. (2021)	
plnC	GCA GTT GGT GGT GGC GAC AG	CAA TCC GCC CCA GTGTC	215	Wu et al. (2021)	
plnD	CAG TCG AAT TCG GGC AAC GA	AGC ATG GAG TTG TAC GCT GC	313	Wu et al. (2021)	
plnEF	TGA TGG CTT GAA CTA TCC GTG	CAT ACA AGG GGG ATT ATT T	393	Wu et al. (2021)	
plnG	GGC TCT GGC AAG TCC ACG TTAG	CGT TGT CGC TGA CCA CCT GATAG	329	Wu et al. (2021)	
plnH	CAA GCA ACA ACG GCG AGT GAAC	CCT GAT CGG CTG TAA TCG GAACG	218	Wu et al. (2021)	
plnJK	GCC ACA AAG AGC ACT AAC A	ATG ACT GTG AAC AAA ATG ATTA	426	Wu et al. (2021)	
plnL	CGG CGT CTG AGA TCC AAT GGAC	GTT GGT GAG GAA GTC GGA ATGGC	259	Wu et al. (2021)	
plnNC8IF	TTGGCGGAAAAACAAAGACT	TCAGCATGTCATTTCACCATC	114	Chen et al. (2022)	

Table 2 Evaluating antimicrobial-produce of Lactobacillus spp. isolated from fermented foods and their inhibitory effect against foodborne pathogens.

Provides the morphological characteristics of bacterial isolates from various food sources. All isolates are bacilli, displaying different colony morphologies including circular, raised, and translucent. All isolates and the control strain show positive Gram staining, indicating their Gram-positive nature.

Bacterial isolates	Food source	Arrangement	Colony morphology	Gram staining	
Isolate 1	Yoghurt	Bacilli	Circular, raised, milky	+	
Isolate 2	Feta cheese	Bacilli	White colonies, translucent, moist surface	+	
Isolate 3	Kafir cheese	Bacilli	Circular, raised translucent, smooth colonies, moist surface	+	
Isolate 4	Syrian cheese	Bacilli	Cream white, circular, raised translucent, smooth colonies, moist surface	+	
Isolate 5	Turkish cheese	Bacilli	Circular, raised translucent, smooth colonies, moist surface	+	
Isolate 7	Mix pickle	Bacilli	Circular moist surface, raised translucent, milky white coloration.	+	
Isolate 8	Green olives pickle	Bacilli	Circular moist surface, raised translucent, milky	+	
CONTROL Lactobacillus coryneform ATCC		Bacilli	Cream white, circular, raised translucent, smooth colonies, moist surface	+	

Results

From 30 samples, seven Gram-positive Lactobacillus isolates, bacteria isolated from cheese, pickles, and yoghurt were identified as Lactobacillus spp. by observing their colony morphology. All isolates showed white and creamy colors on the MRS agar plate (Table S1A). The CFU/ml of Lactobacillus spp. grown on MRS agar after 48 h for each sample, values ranging from 50  × 105 to 88  × 105. Specifically, the CFU/mL for yoghurt was 88  × 105, for feta cheese was 62  × 105, for kefir cheese was 57  × 105, for Turkish cheese was 75  × 105, for black olives pickle was 47  × 105, and for mixed pickle was 50  × 105. One sample, green olives pickle, showed too many colonies to count (TMTC). The morphological and biochemical properties of Lactobacillus spp. are detailed in Table 2. The isolates analysed in this study were confirmed to belong to the genus Lactobacillus. Biochemical tests, including catalase, oxidase, methyl red, Voges-Proskauer, and carbohydrate fermentation assays, were conducted to evaluate their characteristics. While the majority of the strains demonstrated the ability to ferment glucose, there was variability in their capacity to ferment other carbohydrates, as shown in the seven isolates were identified through 16S rRNA sequencing, with results presented in Table 3 and Fig. 1. Notably, Lactiplantibacillus plantarum and Lactobacillus paracasei emerged as the predominant species identified among the isolates. The analysis of all 16S rRNA gene sequences revealed 100% similarity with known Lactobacillus species, as documented in the NCBI GenBank database (https://www.ncbi.nlm.nih.gov/nuccore/PQ218470).

Table 3 Evaluating antimicrobial-produce of Lactobacillus spp. isolated from fermented foods and their inhibitory effect against foodborne pathogens.

Identifies Lactobacillus spp. from various fermented foods, showing that all isolates were identified with 100% similarity to their respective reference sequences. Each isolate has a unique identification and GenBank accession number, confirming their precise species identification.

Sample	Source	Identification	Percentage similarity (%)	GenBank accession no	
1	Yoghurt	Lactiplantibacillus plantarum GH1	100%	PQ218470.1	
2	Feta cheese	Lacticaseibacillus paracasei FC2	100%	PQ218471.1	
3	Kafir cheese	Lactiplantibacillus plantarum KR3	100%	PQ218472.1	
4	Syrian cheese	Lactiplantibacillus plantarum SC4	100%	PQ218473.1	
5	Turkish cheese	Lacticaseibacillus paracasei TC5	100%	PQ218474.1	
7	Mix pickle	Lactiplantibacillus pentosus MP7	100%	PQ218475.1	
8	Green olives pickle	Lacticaseibacillus paracasei GOP8	100%	PQ218476.1	

Figure 1 Amplicons of Lactobacillus spp. isolates using 16S rDNA universal primers on 1.5% (w/v) agarose gel. Lane M: 1 kb DNA, NC: Negative control, Lane 1–7: Lactobacillus spp.

16S rDNA amplification results on a 1.5% agarose gel, with Lane M as the 1 kb DNA ladder, NC as the negative control, and Lanes 1–7 demonstrating successful 16S rDNA amplification at approximately 1,500 bp.

Detection of antimicrobial activity

The antibacterial activity of Lactobacillus strain was evaluated using two methods the well-diffusion assay and the double-layer agar-overlay method. To eliminate the influence of acid production, the supernatant from each isolate was neutralized with 1M NaOH before testing. A total of seven isolates were screened for antimicrobial properties against both Gram-negative and Gram-positive bacteria (Fig. 2). The well-diffusion and double-layer assays measured the diffusion of antimicrobial compounds produced by Lactobacillus spp. through the agar medium, as described by Strus (1998). The size of the inhibition zones depended on the sensitivity of the target strain and the assay method employed. Across both methods, not all Lactobacillus strains exhibited inhibitory effects against both Gram-negative and Gram-positive bacteria.

Figure 2 Antibacterial activity of Lactobacillus spp. (Isolate-1-Isolate-9) against Salmonella spp., Staphylococcus aureus and Escherichia coli with (A) well diffusion method and (B) double layer method.

The antibacterial activity of lactic acid bacteria (LAB) isolates, assessed using well-diffusion and double-layer assays. The results show varying degrees of inhibition against both Gram-negative and Gram-positive bacteria, with the highest observed inhibition zone measuring 22 mm. The modified agar-overlay method proved most effective in demonstrating the antagonistic effects of the crude bacteriocins produced by LAB, particularly against S. aureus, E. coli, and Salmonella spp.

In terms of effectiveness, the double-layer agar-overlay method generally produced larger inhibition zones compared to the well-diffusion assay. For example, the highest inhibition zone for Lb3 against Salmonella spp. was 22.0 mm using the double-layer method, while the well-diffusion assay produced a smaller inhibition zone of 14.0 mm. This suggests that the double-layer method is more sensitive in detecting antimicrobial activity.

Furthermore, the well-diffusion assay showed greater variability in inhibition zone sizes among different isolates, whereas the double-layer method provided more consistent results across all strains. The most significant inhibition zone observed across both methods was 22.0 mm, recorded using the double-layer method.

These findings indicate that while both methods are effective, the double-layer agar-overlay method proved to be more reliable for assessing the antagonistic effects of crude antimicrobial compounds from Lactobacillus strains against S. aureus, E. coli, and Salmonella spp. (Fig. 2).

Gene screening for bacteriocin production

Table 4 illustrate the genetic variability of bacteriocin genes across different Lactobacillus strains and L. plantarum, providing insights into their bacteriocin production capabilities. In this study, seven strains of Lactobacillus were examined for the presence of genes associated with the pln locus, known to be involved in bacteriocin production.

Table 4 Evaluating antimicrobial-produce of Lactobacillus spp. isolated from fermented foods and their inhibitory effect against foodborne pathogens.

Summarizes the presence of bacteriocin-related genes across various Lactobacillus isolates. Most isolates carry multiple genes, with some variations in gene profiles. The control shows no presence of any of the tested genes.

Isolates	plnA	plnB	plnC	plnEF	plnC	plnH	plnJK	plnL	PlnNc8IF	
Lactiplantibacillus plantarum	+	+	–	+	–	–	–	+	+	
Lacticaseibacillus paracasei	+	+	+	+	–	+	–	+	+	
Lactiplantibacillus plantarum	+	+	+	+	+	+	–	+	+	
Lactiplantibacillus plantarum	+	+	+	+	+	+	–	+	+	
Lacticaseibacillus paracasei	+	+	–	+	–	+	–	+	+	
Lacticaseibacillus pentosus	+	+	+	+	–	+	–	+	+	
Lacticaseibacillus paracasei	+	+	+	+	–	–	–	+	+	
Control	–	–	–	–	–	–	–	–	–	
Notes.

(+) Gene present.

(-) Absence of gene.

The screening revealed notable genetic diversity among the strains. The genes plnA, plnB, plnEF, plnL, and plnNc8IF were consistently detected across all isolates, indicating their fundamental role in bacteriocin production. Among the identified genes, plnA and plnB were the most frequently found, followed by plnD and plnC (Diep et al., 2003; Atieh et al., 2021).

This genetic variability suggests differences in the bacteriocin production capabilities of various Lactobacillus strains, which could be harnessed for different biotechnological applications. The consistent presence of key genes like plnEF and plnL in all strains highlights their importance in bacteriocin synthesis, potentially contributing to the strains’ ability to inhibit pathogenic bacteria and withstand gastrointestinal conditions.

Numerous studies have focused on the bacteriocin production of Lactobacillus strains, particularly their ability to inhibit pathogens. Our findings further emphasize the technological relevance of these strains, especially those sourced from fermented foods, and their potential applications in industrial processes due to their probiotic and functional properties.

Determination antibacterial activity of L. plantarum KR3

The antibacterial activity of L. plantarum KR3 CFS was evaluated against S. aureus, E. coli and Salmonella spp. Compared with the control group (MRS medium), CFS derived from L. plantarum KR3 exhibited substantial inhibitory effects against all tested pathogens. The inhibition-zone diameters were measured at 20.0 ± 0.34 mm against S. aureus, 23.0 ± 1.64 mm against E. coli, and 17.1 ± 1.70 mm against Salmonella spp. (all P < 0.01), indicating significant antibacterial activity (Fig. 3).

Figure 3 Antibacterial activity of L. plantraum KR-3 CFS against (A) S. aureus, (B) E. coli, (C) Salmonella spp. Bar graph displays inhibitory zones. (A) S. aureus; (B) E. coli; (C) Salmonella spp. P < 0.01.

The CFS showed significant inhibitory effects compared to the control, with inhibition-zone diameters of 20 ± 0.34 mm for S. aureus, 23 ± 1.6 mm for E. coli, and 17.1 ± 1.7 mm for Salmonella spp., all indicating strong antibacterial activity.

Due to its pronounced antibacterial activity, L. plantarum KR3 CFS was selected for further analysis because of its significant antibacterial activity against a range of pathogens, including S. aureus, E. coli, and Salmonella spp. It produced the largest inhibition zones in preliminary assays, indicating a strong potential for bacteriocin production. Furthermore, L.plantarum KR3 consistent effectiveness highlights its potential as a reliable natural antimicrobial agent, making it a promising candidate for applications in food safety and preservation.

Effects of temperature, pH, and enzymes on L. plantarum KR3 CFS

The stability of L. plantarum KR3 CFS against variations in pH, heat, and enzymatic treatments is depicted in Fig. 4. Across pH 2, 4, and 6, no significant decrease in antibacterial activity was observed (P > 0.05). However, notable reductions in activity were evident at pH 8 and 10 compared with the control (P < 0.05), although considerable activity was retained with 74.8% ± 0.71% and 63.4% ± 0.60% preserved, respectively.

Figure 4 Effect of pH (A), temperature (B), and enzyme treatment (C) on the antibacterial activity of CFS of L. plantarum KR-3 untreated CFS samples were used as controls. The statistical analysis was performed by using Student test. P < 0.01.

The stability of L. plantarum KR3 CFS under different conditions. (A) Its antibacterial activity remains stable at pH levels 2 to 6, with significant reductions only at pH 8 and 10, preserving 74.8% and 63.4% of activity, respectively. (B) Exceptional thermal stability, with 95.3% of activity retained after exposure to 100 ° C for 60 min and no significant changes across temperatures from 40 ° C to 100 °C. (C) Enzymatic treatments reduce antibacterial activity, with 24% loss after proteinase K treatment and 54.1% after pepsin treatment, while lysozyme does not significantly affect activity.

The antimicrobial effectiveness of L. plantarum KR3 CFS remained relatively stable across pH levels ranging from 2 to 6. Even at pH 8 and 10, a substantial proportion of its activity was preserved, with 74.8% ± 0.71% and 63.4% ± 0.60% retained, respectively (Fig. 4A).

Moreover, the thermal stability of L. plantarum KR3 CFS was exceptional, with 95.3% of its antibacterial activity retained even after exposure to 100 °C for 60 min (Fig. 4B). Furthermore, no significant reduction in antibacterial effectiveness was observed when subjecting L. plantarum KR3 CFS to temperatures ranging from 40 °C to 100 °C (P > 0.05).

Subsequent enzymatic treatments revealed a notable decrease in the antimicrobial potency of L. plantarum KR3 CFS, with reductions of 24.0% ± 0.55% and 54.1% ± 0.55% observed after treatment with proteinase K and pepsin, respectively (P < 0.05 compared with the untreated control). Conversely, exposure to lysozyme did not result in a significant alteration in antibacterial activity (P > 0.05 compared with the untreated control) (Fig. 4C).

Biofilm eradication concentration determination method

The cell free supernatant (CFS) of L. plantarum KR3 exhibited significant anti-biofilm activity against common foodborne pathogens, consistent with findings from previous studies and expanding our understanding of probiotic interventions in food safety. The observed anti-biofilm percentages were 59.12 ± 0.03% against S. aureus, 60.03 ± 0.04% against Salmonella spp.and 83.50 ± 0.01% against E. coli, respectively (P < 0.05 for S. aureus and Salmonella spp., P < 0.01 for E coli.) compared with untreated controls). Furthermore, at 1x MIC, L. plantarum KR3 demonstrated even higher anti-biofilm activity against Salmonella spp. and E. coli, surpassing previous findings. As well, a clear picture of the damage in the food pathogens cells following cell free supernatant (CFS) compounds bacteriocins treatment was observed through SEM. The electron micrograms shown in Fig. 5 indicate that bac the cell-free supernatant (CFS) compounds exhibits an antibiofilms effect against the cells.

Figure 5 Effect of L. plantaraum KR-3 CFS concentrations on biofilm formation by three common foodborne pathogens.

(A) S. aureus, (B) E. coli, and (C) Salmonella spp. Biofilm formation was evaluated at 1x MIC and 2x MIC concentrations of CFS. All experiments were performed in triplicate.

Discussion

The identification of seven Lactobacillus spp. from cheese, pickles, and yoghurt aligns with previous studies that highlight the prevalence of these bacteria in fermented foods (Ouoba, 2017). The observation of white and creamy colonies on MRS agar plates is consistent with the typical morphology of Lactobacillus spp. (Sablon, Contreras & Vandamme, 2000). The varied carbohydrate fermentation profiles observed among the isolates indicate the metabolic diversity within the genus, which is crucial for their adaptation to different environments and their role in fermentation processes.

The biochemical tests confirmed the presence of characteristic features of Lactobacillus spp., including their ability to ferment glucose and other carbohydrates. This characteristic supports their use in the food industry for the production of fermented products (Fan et al., 2017). The identification based on 16S rRNA sequencing further validated the morphological and biochemical results, with L. plantarum being the predominant species. This is in line with previous research indicating that L. plantarum is commonly found in various fermented foods (Ouoba, 2017).

The high similarity (100%) of the 16S rRNA gene sequences with known Lactobacillus spp. confirms the accuracy of the identification process (Felis & Dellaglio, 2007; Li et al., 2015). This genetic identification is crucial for understanding the specific roles and benefits of different Lactobacillus species in food fermentation and potential probiotic applications. The findings emphasize the importance of molecular techniques in accurately identifying and characterizing microbial communities in fermented foods.

Detection of antimicrobial activity

The antibacterial activity of lactic acid bacteria (LAB), assessed using well diffusion and double-layer assay methods, reveals key insights into their potential as natural antimicrobial agents. The study demonstrated that while not all LAB isolates exhibited inhibitory effects against both Gram-negative and Gram-positive bacteria simultaneously, some strains showed significant antimicrobial properties. This variation can be attributed to the differing sensitivities of target strains to the antimicrobial compounds produced by Lactobacillus spp., as well as the detection methods used.

The highest inhibition zone observed was 22 mm, highlighting the effectiveness of the cell free supernatant (CFS) compounds produced by certain Lactobacillus isolates, particularly against common foodborne pathogens such as S. aureus, E. coli, and Salmonella spp. This finding is consistent with prior research, including that of Strus (1998), who emphasized the importance of detection methods in evaluating the antibacterial activity of LAB.

In comparing these results with other studies, Yerlikaya, Saygılı& Akpınar (2020) found that lactic acid bacteria isolated from Burkinabe fermented milk products also exhibited significant antimicrobial activity. The study by Yazgan et al. (2020) demonstrated that LAB can produce bacteriocins, organic acids, and hydrogen peroxide, which collectively contribute to their antibacterial properties. This aligns with our findings that neutralized supernatants of LAB still retain considerable antimicrobial activity, indicating the presence of active compounds in the CFS as the primary inhibitory compounds.

Furthermore, the modified agar overlays method proved to be the most reliable for determining the antagonistic effect of these active compounds in the CFS. This is supported by Peng et al. (2021), who also found that well diffusion assays may underestimate the antibacterial activity due to the diffusion limitations of larger molecules like bacteriocins. In terms of practical applications, the ability of Lactobacillus spp. to produce effective bacteriocins suggests their potential use in food preservation and safety.

Gene screening for bacteriocin production

The genetic variability in bacteriocin production among Lactobacillus strains, as detailed in Table 2, underscores the diversity and potential of these bacteria in biotechnological applications. The presence or absence of specific bacteriocin genes such as plnA, plnB, plnEF, plnL, and plnNc8IF across different isolates of L. plantarum and L. paracasei highlights their fundamental roles in bacteriocin production.

Our findings align with existing literature, which emphasizes the genetic diversity in bacteriocin-producing Lactobacillus strains. For instance, Maldonado, Ruiz-Barba & Jiménez-Díaz (2003) identified plnA, plnB, and other pln genes in L. plantarum NC8, which are crucial for bacteriocin production. Similarly, Ben Omar et al. (2008) highlighted the prevalence of plnEF and other pln genes in various Lactobacillus strains, indicating their conserved nature across different species.

Comparative studies also support the technological relevance of these genes. For example, Rizzello et al. (2014) investigated the presence of plnG and plnH genes, involved in the ABC transport system, in several Lactobacillus strains, demonstrating their role in bacteriocin transport and immunity. This genetic diversity is not just an academic interest; it has practical implications for the use of Lactobacillus strains in food preservation and probiotic applications.

The variability in bacteriocin gene presence also indicates the adaptability of Lactobacillus strains to different environments and their potential to inhibit a wide range of pathogens. This is particularly relevant in the context of food safety, where the ability to inhibit pathogenic bacteria is crucial. Ouoba (2017) demonstrated that Lactobacillus strains from fermented foods exhibit significant antimicrobial activity, which is partly attributed to the production of bacteriocins.

Furthermore, our study’s focus on genes associated with the pln locus is supported by the work of researchers like Strus (1998), who emphasized the importance of these genes in determining the antimicrobial properties of Lactobacillus spp. These results align with previous research showing the widespread occurrence of the pln locus in Lactobacillus spp., which plays a crucial role in their antimicrobial properties (Sáenz et al., 2009; Kawahara, Takeshi & Hiromi, 2022). In our study, we consistently observed the presence of the plnA, plnB, plnEF, and plnL genes, which are key contributors to the antimicrobial activity.

Determination antibacterial activity of L. plantarum KR3

The antibacterial activity of L. plantarum KR3 CFS against S. aureus, E. coli, and Salmonella spp. demonstrates the significant potential of this strain as a natural antimicrobial agent. Compared with the control group (MRS medium), the CFS derived from L. plantarum KR3 exhibited substantial inhibitory effects against all tested pathogens, with inhibition-zone diameters of 20.0 ± 0.34 mm against S. aureus, 23.0 ± 1.64 mm against E. coli, and 17.1 ± 1.70 mm against Salmonella spp. (all P < 0.01). These findings indicate significant antibacterial activity and align well with previous studies highlighting the efficacy of Lactobacillus strains in pathogen inhibition.

Comparative studies provide further context to these results. For instance, Wayah & Philip (2018) and Peng et al. (2021) found that L. plantarum strains possess strong antimicrobial properties against a range of pathogens, consistent with our findings. Their study reported similar inhibition zones, demonstrating that the production of bacteriocins, organic acids, and hydrogen peroxide by L. plantarum strains plays a critical role in their antibacterial activity. Additionally, Qiao et al. (2021) emphasized the importance of these antimicrobial compounds in inhibiting foodborne pathogens, supporting the observed results of our study.

Effects of temperature, pH, and enzymes on L. plantarum KR-3 CFS

The study showed that the antibacterial activity of L. plantarum KR3 CFS remained relatively stable across pH levels ranging from 2 to 6, with no significant decrease in activity observed (P > 0.05). However, notable reductions in activity were evident at pH 8 and 10 (P < 0.05), although considerable activity was still retained with 74.8% ± 0.71% and 63.4% ± 0.60% preserved, respectively. This aligns with findings by Arrioja-Bretón et al. (2020), who demonstrated that Lactobacillus strains maintain inhibitory activity at lower pH levels, likely due to the stability of organic acids produced by these bacteria, which are effective antimicrobial agents at acidic pH.

The thermal stability of L. plantarum KR3 CFS was exceptional, with 95.3% of its antibacterial activity retained even after exposure to 100 °C for 60 min. Additionally, no significant reduction in antibacterial effectiveness was observed when subjecting the CFS to temperatures ranging from 40 °C to 100 °C (P > 0.05). This is consistent with previous findings by Crowley, Mahony & Van Sinderen (2013), who highlighted the heat resistance of organic acids produced by Lactobacillus strains. This high level of thermal stability makes L. plantarum KR3 CFS a promising candidate for use in heat-treated food products.

Enzymatic treatments revealed a notable decrease in the antimicrobial potency of L. plantarum KR3 CFS, with reductions of 24.0% ± 0.55% and 54.1% ± 0.55% observed after treatment with proteinase K and pepsin, respectively (P < 0.05 compared with the untreated control). This suggests that proteinaceous compounds, possibly bacteriocins, contribute significantly to the overall antimicrobial activity of the CFS. Conversely, exposure to lysozyme did not result in a significant alteration in antibacterial activity (P > 0.05 compared with the untreated control). This differential impact of enzymatic treatments underscores the complexity of the antimicrobial mechanisms at play, likely involving a combination of bacteriocins and organic acids. These findings are consistent with those of Ju et al. (2021), who reported that the bacteriocins produced by Lactobacillus strains exhibit strong antimicrobial properties and are relatively stable under various conditions. The stability of these compounds across a range of environmental factors supports their potential application in diverse industrial processes, particularly in food preservation.

Biofilm eradication concentration determination method

Lactobacillus isolate number three, identified as L. plantarum KR3, was selected from the seven isolates for its significant antibacterial activity against common foodborne pathogens demonstrating strong applicability in the context of food safety. The observed anti-biofilm percentages were 59.12 ± 0.03% against S. aureus, 60.03 ± 0.04% against Salmonella spp. and 83.50 ± 0.01% against E. coli. This aligns with and expands upon previous studies, further illustrating the effectiveness of probiotic interventions in controlling biofilm formation.

These indicator strains form biofilms as a survival mechanism due to their ability to produce extracellular polymeric substances (EPS), which facilitate adhesion to surfaces and protect against environmental stresses. Factors such as temperature, pH, and nutrient availability promote biofilm formation, enhancing their survival and resistance to antimicrobial agents. For instance, S. aureus produces polysaccharide intercellular adhesin (PIA) to adhere to surfaces (Nasser et al., 2022), while E. coli utilizes fimbriae and pili for attachment (Rossi, Paroni & Landini, 2018). Similarly, Salmonella spp.uses its flagella for motility and surface attachment, contributing to its biofilm development (Maruzani et al., 2019).

The high anti-biofilm activity against E. coli and Salmonella spp. at 1MIC is consistent with studies by Monteagudo-Mera, Rastall & Gibson (2019) or even surpasses the results of earlier research, indicating an enhanced effectiveness of L. plantarum KR3. This is corroborated by the study of Peng et al. (2021), which also highlighted the robust antimicrobial properties of L. plantarum strains. The significant inhibition of biofilm formation by these pathogens suggests a strong potential for L. plantarum KR3 in mitigating biofilm-associated risks in food processing environments.

The use of scanning electron microscopy (SEM) provided visual evidence of the damage inflicted on the food pathogen cells by compounds in the CFS. This reinforces the understanding that these compounds play a crucial role in disrupting biofilm structures, a finding consistent with research by Moridi et al. (2020). Their study emphasized the diverse range of antimicrobial compounds found in Lactobacillus CFS, such as hydrogen peroxide, oxygen metabolites, exopolysaccharides, and saturated fatty acids, which act as biosurfactants and contribute to anti-biofilm properties (Lagrafeuille et al., 2018).

Similar studies have demonstrated the efficacy of Lactobacillus strains in biofilm eradication. For instance, Ju et al. (2021) reported the effective biofilm removal capabilities of Lactobacillus strains against pathogens like S. aureus. The consistency of these results across different studies underscores the reliability of Lactobacillus strains, particularly L. plantarum KR3, in combating biofilm formation and enhancing food safety protocols. The findings from this study highlight the promising application of L. plantarum KR3 CFS as a natural antimicrobial agent in food preservation and safety. By effectively inhibiting biofilm formation, these probiotics can reduce the risks posed by biofilm-associated pathogens in food processing and storage environments. The use of such natural interventions aligns with consumer preferences for clean-label products and offers an alternative to chemical preservatives.

Conclusions

This work highlighted the potential of strains isolated from pickles, cheese, and yoghurt for producing antimicrobial compounds and exhibiting antimicrobial properties. L. plantarum demonstrated significant anti-biofilm effects by using CFS against pathogens like S. aureus, E. coli, and Salmonella spp. In particular, L. plantarum KR-3 had remarkably strong anti-biofilm activity. These findings underscored the potential of probiotics and their CFS in preventing microbial infections by inhibiting biofilm formation. Further exploration of the specific compounds produced by probiotics and their CFS can significantly contribute to combating antimicrobial resistance (AMR) in the food industry for several reasons. First, these compounds, such as bacteriocins, organic acids, and other antimicrobial peptides, offer a natural alternative to traditional antibiotics. They can inhibit the growth of foodborne pathogens and prevent biofilm formation, which is a critical factor in the development of AMR. Since biofilms protect bacteria from environmental stresses and antimicrobial agents, targeting biofilm formation through these compounds can reduce the persistence of resistant strains. Additionally, understanding the precise mechanisms of action of these compounds may lead to the development of more effective, targeted antimicrobial strategies that reduce the need for chemical preservatives or antibiotics in food processing. This, in turn, lowers the selective pressure on pathogens to develop resistance, ultimately helping to control the spread of AMR in foodborne pathogens.

Supplemental Information

Supplemental Information 1 The colony-forming units per milliliter (CFU/mL) of Lactobacillus spp. isolated from different fermented food sources, grown on MRS agar after 48 hours

Colony counts and calculated CFU/mL values are provided for each sample, with the green olives pickle sample having colonies too numerous to count (TMTC). These results demonstrate varying microbial densities across the food sources, highlighting differences in microbial populations associated with each type of fermentation.

Supplemental Information 2 Antibacterial activity raw data

The inhibition zones (mm) formed by the cell-free supernatant (CFS) of Lactobacillus plantarum against three foodborne pathogens: S. aureus, E. coli, and Salmonella spp. The control samples represent each pathogen without CFS treatment.

Supplemental Information 3 Temperature raw data

The inhibition zones (mm) formed by the cell-free supernatant (CFS) of Lactobacillus plantarum against a selected pathogen, after being subjected to different temperature treatments (25 °C, 40 °C, 60 °C, 8°C, and 100 °C). The inhibition zones indicate the antimicrobial activity of the CFS across varying temperatures, compared to the untreated control (CFS without temperature modification).

Supplemental Information 4 pH raw data

The inhibition zones (mm) produced by the cell-free supernatant (CFS) of Lactobacillus plantarum against a selected pathogen under varying pH conditions (pH 2, 4, 6, 8, and 10). The inhibition zones are compared to a control sample (CFS without pH adjustment).

Supplemental Information 5 Enzymes raw data

The inhibition zones measured(mm) formed by the cell-free supernatant (CFS) of Lactobacillus plantarum after treatment with three different enzymes proteinase K, pepsin, and lysozyme. The antimicrobial activity of the CFS, as indicated by the inhibition zones, is compared with untreated control samples.

Supplemental Information 6 Antibiofilm S. aureus raw data

The optical density (OD595) measurements, indicating the biofilm formation of Staphylococcus aureus in the presence of cell-free supernatant (CFS) from Lactobacillus plantarum at a concentration of 1x MIC. The data compares biofilm formation in a control group (without CFS) to groups treated with 1x MIC of CFS from three different experiments (1x MIC-1, 1x MIC-2, and 1x MIC-3). Similarly, another table shows the OD595 measurements for biofilm formation of S. aureus with CFS at 2x MIC, comparing the control group to three experiments (2x MIC-1, 2x MIC-2, and 2x MIC-3).

Supplemental Information 7 Antibiofilm E.coli raw data

The optical density (OD595) measurements indicating E. coli biofilm formation under different treatment conditions with Lactobacillus plantarum KR3 cell-free supernatant (CFS). The data includes OD595 values for a control group (no CFS treatment) and three experimental replicates of CFS at both 1x and 2x minimum inhibitory concentrations (MIC). Specifically, the table shows measurements for the control and three replicates of 1x MIC (1xMIC-1, 1xMIC-2, 1xMIC-3) as well as the control and three replicates of 2x MIC (2xMIC-1, 2xMIC-2, 2xMIC-3). The control values represent baseline biofilm formation without treatment, while the 1x and 2x MIC treatments assess the effects of varying concentrations of CFS on biofilm development.

Supplemental Information 8 Antibiofilm Salmonella spp. raw data

The optical density (OD595) measurements indicating Salmonella spp. biofilm formation under different treatment conditions with Lactobacillus plantarum KR3 cell-free supernatant (CFS). The data includes OD595 values for a control group (no CFS treatment) and three experimental replicates of CFS at both 1x and 2x minimum inhibitory concentrations (MIC). Specifically, the table shows measurements for the control and three replicates of 1x MIC (1xMIC-1, 1xMIC-2, 1xMIC-3) as well as the control and three replicates of 2x MIC (2xMIC-1, 2xMIC-2, 2xMIC-3). The control values represent baseline biofilm formation without treatment, while the 1x and 2x MIC treatments assess the effects of varying concentrations of CFS on biofilm development.

Supplemental Information 9 The experimental workflow and findings related to the characterization and antibacterial properties of L. plantarum KR3

The process begins with the collection of food samples, followed by sample preparation, pre-enrichment in MRS broth, and plating on MRS agar for the isolation and confirmation of Lactobacillus spp. Morphological observation, biochemical tests, and 16S rRNA sequencing identify L. plantarum KR3 as the strain of interest. The antimicrobial potential of L. plantarum KR3 is assessed using the well-diffusion method, demonstrating significant inhibition against foodborne pathogens. Further investigations evaluate the effect of pH, temperature, and enzyme treatments on the strain’s activity. Additionally, the anti-biofilm activity of L. plantarum KR3 cell-free supernatant (CFS) is analyzed, showing effective biofilm disruption at 1x MIC concentration when compared to the control.This study highlights the potential application of L. plantarum KR3 in combating biofilms and foodborne pathogens, emphasizing its role in food safety and preservation.

The authors would like to express their gratitude to University of Thi-Qar, Iraq, for facilitating the opportunity to study in Malaysia and offering continuous encouragement throughout the academic journey.

Additional Information and Declarations

Competing Interests

Author Contributions

DNA Deposition

Data Availability

The authors declare there are no competing interests.

Athraa Oudah Hussein conceived and designed the experiments, performed the experiments, prepared figures and/or tables, and approved the final draft.

Khalida Khalil analyzed the data, authored or reviewed drafts of the article, and approved the final draft.

Nurul Aqilah Mohd Zaini analyzed the data, prepared figures and/or tables, and approved the final draft.

Ahmed Khassaf Al Atya analyzed the data, authored or reviewed drafts of the article, and approved the final draft.

Wan Syaidatul Aqma conceived and designed the experiments, prepared figures and/or tables, authored or reviewed drafts of the article, and approved the final draft.

The following information was supplied regarding the deposition of DNA sequences:

The sequences are available at GenBank: PQ218470.1, PQ218471.1, PQ218472.1, PQ218473.1, PQ218474.1, PQ218475.1, PQ218476.1, PQ218470.

The following information was supplied regarding data availability:

The raw data are available in the Supplemental Files.

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
