# Peer review of "Antimicrobial activity of Lactobacillus spp. isolated from fermented foods and their inhibitory effect against foodborne pathogens"

_PeerJ, doi:10.7717/peerj.18541_

## Round 0.1 · original submission · Major Revisions

Dear authors, I ask you to carefully respond to each of the reviewers' comments. This will allow you to bring your manuscript as close as possible to modern requirements for scientific publications. I hope that this manuscript will become a serious step in the study of the practically important genus of bacteria from the genus Lactobacillus.

Reviewer 1 ·

Basic reporting

This work deal with antimicrobial activity of L. plantarum strains isolated from different source. The topic is interesting and on time. However, there are some sentences or part of the text that are bit unclear/confused. For example, Materials and method section should be improved and must be linear. The strains nomenclature (e.g., L. plantarum KR3) should be reported in tables/figures as well.

Experimental design

Materials and method
Line 66-74
- The number of strains for analysed products may be reported.
- Please, specify if the identification was performed on all the strains isolated or only on randomly picked colony.
- The 16S gene, at least for the most important strain identified, is usualy submitted to a Genebank.
- Please, is unclear when the CFS was collected (growth phase) and if the growth curve was similar or different for the "active" strains analysed. In the analysis of bacteriocin production, an enzymatic tretment of CFS is apparently missing (see line 76-79). It is reported later with pH and temperature effect as well. Should be included in this part.
- The origin of the indicator strains (E. coli, S. aureus, and Salmonella spp.) should be reported. Animal or vegetal origin? Any reference? Where these strains come from? Collection?

Validity of the findings

-Line 174/176. At this stage is too early to talk about bacteriocine....missing the enzymatich treatment for "potential" bacteriocine.
- 163/198. This part is rather than linear. Please rewrite. Please, specify why L. plantarum KR3 CFS was chosen for further analysis.

- Unclear the meaning "....from Lactobacillus isolate number three, particularly L. plantarum KR3, ..."
(line 346-347).

In general
The identification of genes codifying for bacteriocine does not mean that the strain is able to produce it. Bacteriocine should be isolated and the AA structure identified.

Additional comments

Please, consider the new nomenclature of L. plantarum and use Lactiplantibacillus plantarum rather than Lactiplantibacillus Plantarum (see table 4)

Reviewer 2 ·

Basic reporting

The authors conducted a large amount of research on identification and the antimicrobial activity of Lactobacillus strains isolated by them. The obtained results are important for revealing the potential of probiotics with antimicrobial properties that produce bacteriocins and prevent microbial infections by inhibiting biofilm formation.
After the first mention in the text (abstract, introduction), the generic names of the strains should be written in full (lines 27, 54). Strains or isolates Lactobacillus (in the entire text), members of the genus Bacillus (line 236) should be highlighted in italic.
Digital data of inhibition zones and antibacterial activity must be submitted with the same digits after the point for mean values and standard deviation values.
The English language should be improved to ensure that an international audience can clearly understand your text. Some examples where the language could be improved include lines 84 (…were cultured or cultivated?), 95 (…(3) indicator pathogenic bacteria … were added to (5 mL) of MHA…), 133 (centrifuged at 7,104 × g), 222, 298 – the current phrasing makes comprehension difficult.
References in text should be given in the form: author et al., year (in the entire text). The list does not include links to: Aijuan Wu et al., 2021; Jun Chen et al., 2020 (table 1), Macwana and Muriana, 2012 (line 107), Li et al., 2009 (line 244). There are no references in the text to: Hurtado et al., 2011 (line 404), Monteagudo-Mera et al., 2019 (line 426), Nataro and Kaper (line 436), Randazzo et al., 2002 (line 452), Rocha-Ramírez et al., 2023 (line 455), Sablone et al., 2000 (line 462), Sunita et al., 2012 (line 467).
Jiang et al. (2022a) (line 120) or Jiang et al. (2022)? Moradi (line 360) or Moridi (line 429)?
All references must be formatted according to the journal’s requirements.
Fig. 1: Lactobacillus spp.
Fig. 2: errors in the signature of the x-axis (a and b).
Fig. 5: errors in the signatures of the x-axis and y-axis (a, b, c). Signatures must be in capital letters.
Table 1: errors in the table title and references.
Table 4: species names are written with a lowercase letter.

Experimental design

Please describe in more detail how the сell-free supernatant extracts were prepared and how the cells were destroyed (lines 78-79).
Describe which collection the indicator strains E. coli, S. aureus, and Salmonella spp. are from, provide numbers and references (lines 83-84).
Why do you think that the indicator strains form biofilm (line 122)? Please provide literature references.

Validity of the findings

You write that (line 188): “We examined nine Lactobacillus strains for the presence and expression of genes associated with the pln locus, known for their involvement in bacteriocin production”. Provide literature references, please.
Explain in more detail, please, why further exploration of the specific compounds (lines 381-382) can lead to combating antimicrobial resistance in the food industry?

Additional comments

The article can be published after revision.

Annotated reviews are not available for download in order to protect the identity of reviewers who chose to remain anonymous.

Reviewer 3 ·

Basic reporting

- Clear, unambiguous, professional English language used throughout – Yes.
- Intro & background to show context. Literature well referenced & relevant – Yes.
- Structure conforms to PeerJ standards, discipline norm, or improved for clarity – Yes.
- Figures are relevant, high quality, well labelled & described – Partially (see section 4).
- Raw data supplied – Yes.

Experimental design

- Original primary research within Scope of the journal - Yes.
- Research question well defined, relevant & meaningful. It is stated how the research fills an identified knowledge gap – Partially (see section 4).
- Rigorous investigation performed to a high technical & ethical standard – Partially (see section 4).
- Methods described with sufficient detail & information to replicate – Partially (see section 4).

Validity of the findings

Results are interesting, because problem of foodborne pathogens is very significant and need modern “ecodecision”.
- All underlying data have been provided; they are robust, statistically sound, & controlled – Yes.
- Conclusions are well stated, linked to original research question & limited to supporting results – Partially (see section 4).

Additional comments

1. In Title:
- term “Evaluating” can be reject from title.
2. In Abstract:
- in: Culture supernatants from nine Lactobacillus isolates showed varying inhibitory activities “Lactobacillus” is optimal to change on “lactobacilli”.
- about including into probiotics: if was study only culture supernatants is too early to make same decision.
3. In Introduction:
- there is no aim of the research.
4. In Materials and Methods:
- line 72: Genomic DNA was extracted using a DNA extraction kit. What kind of kit?
- line 73: what amplifier was used? And what kind of reactives?
- line 83: characteristics of indicator stains: collection, seral number? Or it were clinical strains?
- line 85: “Petri” must be writing from capital letter.
- Well-diffusion method and Overlay-plate method are partially corresponding with bacteriocines synthesis study. It’s study of antagonistic activity of strains, that can be realized by very many different compounds, not only bacteriocines. Please, clear this issue in chapter and don’t unite b and c in “Screening of Lactobacillus spp. for bacteriocin production”. Optimally to change this sentence like “Screening of Lactobacillus spp. for antagonistic activity”.
- what is the criteria of biofilm formation and eradication? What is control?
5. In Results:
- fig. 1a and 1b are not obligatory.
- it’s no need to list the typical biochemical tests that were used for lactobacilli identification in this chapter.
- table a is not obligatory.
- in Materials and Methods authors wrote 1 M NaOH. In Results – 1 N NaOH. Please, unify.
- fig. 2: what are the difference between a and b? Please, clarify it in text.
- fig. 3 is not obligatory.
- line 180: if authors indicate here L. paracasei, they must wrote about it above in text, but there are no any information about it identification.
- effects of pH: naturally that in pH 8 and more take place inhibition, because acids are neutralized. Decrease of antimicrobial activity in these condition can be explaine by neutralizing of acids. In this case appear a question: is antimicrobial activity of lactobacilli strains realized by bacteriocines or due to lactic acid production? For support of bacteriocines role it need to do mRNA analysis, determine not only presence of gene, but also it expression.
- fig. 6: please, describe what are a, b and c?
- line 266-367: “The electron micrograms shown in Fig 6 indicate that bacteriocins exhibits an antibiofilms effect against the cells”. There are no any proofs that is a bacteriocin effect. It’s effect of supernatant. And will enough “antibiofilm effect” without “against the cells”, because biofilm is not only cells.
6. In Discussion:
- line 236-237: for what in text include information about genus Bacillus and it possibility to formation spores?
7. In Conclusion:
- from where authors received information about L. pentosus, if there are no any information about it in text?

Annotated reviews are not available for download in order to protect the identity of reviewers who chose to remain anonymous.

---

## Round 0.2 · accepted · Accept

Dear authors, I am pleased to inform you that all three reviewers have approved your publication. I recommend your article for publication and hope that you will continue your research in this direction. I believe that your research will help improve the quality of fermented foods and increase the biological value of the human diet. Good luck in your future scientific work.

Reviewer 1 ·

Basic reporting

Authors correctly answered to my queries

Experimental design

Improved

Validity of the findings

Ok

Additional comments

Paper has been improved

Reviewer 2 ·

Basic reporting

Thanks to the authors for taking into account all comments and making changes to the manuscript.

Experimental design

No comment.

Validity of the findings

No comment.

Additional comments

The article can be published after grammatical editing of the text.

Reviewer 3 ·

Basic reporting

-

Experimental design

-

Validity of the findings

-

Additional comments

-